# Gene therapy for aromatic L-amino acid decarboxylase deficiency by MR-guided direct delivery of AAV2-AADC to midbrain dopaminergic neurons

Toni S. Pearson [1,2,12], Nalin Gupta [1,12], Waldy San Sebastian[1], Jill Imamura-Ching[1], Amy Viehoever [3], Ana Grijalvo-Perez[3], Alex J. Fay[3], Neha Seth[4], Shannon M. Lundy[5], Youngho Seo [6], Miguel Pampaloni[6], Keith Hyland[7], Erin Smith[8], Gardenia de Oliveira Barbosa[9], Jill C. Heathcock[9], Amy Minnema[10], Russell Lonser[10], J. Bradley Elder[10], Jeffrey Leonard[10,11], Paul Larson[1] & Krystof S. Bankiewicz[1,10✉]

Aromatic L-amino acid decarboxylase (AADC) deficiency is a rare genetic disorder characterized by deficient synthesis of dopamine and serotonin. It presents in early infancy, and causes severe developmental disability and lifelong motor, behavioral, and autonomic symptoms including oculogyric crises (OGC), sleep disorder, and mood disturbance. We investigated the safety and efficacy of delivery of a viral vector expressing AADC (AAV2-hAADC) to the midbrain in children with AADC deficiency (ClinicalTrials.gov Identifier NCT02852213). Seven (7) children, aged 4–9 years underwent convection-enhanced delivery (CED) of AAV2-hAADC to the bilateral substantia nigra (SN) and ventral tegmental area (VTA) (total infusion volume: 80 µL per hemisphere) in 2 dose cohorts: $1.3 \times 10^{11}$ vg (n = 3), and $4.2 \times 10^{11}$ vg (n = 4). Primary aims were to demonstrate the safety of the procedure and document biomarker evidence of restoration of brain AADC activity. Secondary aims were to assess clinical improvement in symptoms and motor function. Direct bilateral infusion of AAV2-hAADC was safe, well-tolerated and achieved target coverage of 98% and 70% of the SN and VTA, respectively. Dopamine metabolism was increased in all subjects and FDOPA uptake was enhanced within the midbrain and the striatum. OGC resolved completely in 6 of 7 subjects by Month 3 post-surgery. Twelve (12) months after surgery, 6/7 subjects gained normal head control and 4/7 could sit independently. At 18 months, 2 subjects could walk with 2-hand support. Both the primary and secondary endpoints of the study were met. Midbrain gene delivery in children with AADC deficiency is feasible and safe, and leads to clinical improvements in symptoms and motor function.

A full list of author affiliations appears at the end of the paper.

Aromatic L-amino acid decarboxylase (AADC) deficiency is a rare autosomal recessive neurodevelopmental disorder that typically presents in infancy, and is characterized biochemically by deficiency of the catecholamines (dopamine, epinephrine, and norepinephrine) and serotonin[1]. Approximately 135 patients with this condition have been reported in the literature worldwide, though the true incidence is unknown[2]. The AADC enzyme catalyzes the synthesis of dopamine and serotonin from their respective precursors. Children with AADC deficiency cannot metabolize levodopa to dopamine, and for this reason, patients typically do not respond to treatment with exogenous levodopa. In addition, they derive little or no benefit from other medical therapies[2,3]. The majority of patients experience severe long-term physical and intellectual disability, and are at risk of premature death from complications related to the disease.

AADC deficiency causes a variety of motor, autonomic, and behavioral symptoms. The motor symptoms include hypokinesia, hypotonia, dystonia, and oculogyric crises (OGC). OGC are one of the cardinal features of the disease, and are characterized by episodes of intermittent or sustained tonic vertical (usually upward), horizontal, or convergent deviation of the eyes, sometimes accompanied by dystonia or other involuntary movements of the face and body. Episodes typically last for hours and occur several times per week[4]. Catecholamine and serotonin deficiency also cause non-motor symptoms such as emotional lability, sleep disturbance, and excessive sweating. Moderate to severe motor and intellectual disability is present in at least 70% of the patients described to date[2,4,5].

A number of surgical strategies have been used to treat adult patients with Parkinson's disease (PD), a neurodegenerative disorder in which brain AADC activity is progressively lost due to disease progression or cell death. These include placement of deep brain stimulation electrodes[6], cell transplants[7], and gene therapy[8]. Most clinical trials evaluating the efficacy of gene therapy for PD have utilized an adeno-associated virus type 2 (AAV2), encoding a neurotrophic factor[9,10] or the *DDC* gene (hAADC), delivered by a direct surgical procedure to the putamen[11,12]. The rationale for selection of the putamen as the anatomic target in PD for *DDC* gene therapy is that transduction of the post-synaptic cells that express dopamine receptors in the putamen may provide sufficient AADC activity to cause an increase in local metabolism of exogenous levodopa (i.e., dopaminergic medication) and a subsequent increase in striatal levels of dopamine[13].

In contrast to previous studies in PD, we hypothesized that patients with AADC deficiency would benefit from AAV2-hAADC delivery to two specific regions of the midbrain: the substantia nigra pars compacta (SNc) and the ventral tegmental area (VTA). The rationale for the selection of the midbrain target as opposed to the putamen is that dopaminergic neurons in the midbrain and their axonal projections are structurally intact in children with AADC deficiency[14]. Furthermore, the mood and autonomic symptoms that accompany severe motor impairment may be attributable in part to dopamine deficiency beyond the nigrostriatal system, and thus would not be treated by putaminal transduction.

AAV2-hAADC gene delivery to the putamen for children with AADC deficiency has been investigated in Taiwan[15,16] and Japan[17]. In all, 14 subjects in Taiwan (ages 1.7–8.4 years, total dose $1.5 \times 10^{11}$ vg in 160 μL[15] or $1.81 \times 10^{11}$ vg in 160 μL[16]) and 6 subjects in Japan (ages 4–19, total dose $2 \times 10^{11}$ vg in 200 μL[17]) were treated with bilateral intraputaminal infusion of AAV2-hAADC. The authors reported some clinical improvements, but OGC persisted in all subjects. Only 2 of the 12 patients in the latter two studies who had been followed for at least 2 years gained the ability to sit independently[16,17].

By delivering AAV2-hAADC to the SNc and VTA, our goal was to increase AADC enzyme activity in midbrain dopaminergic neurons, thereby rescuing dopamine biosynthesis and dopaminergic neurotransmission in nigrostriatal, mesolimbic, and mesocortical pathways. Midbrain delivery also takes advantage of anterograde axonal transport of AAV2 from these regions to deliver AAV2-hAADC to neuroanatomically appropriate brain regions like the striatum[18,19]. To maximize safety and targeting accuracy of the gene vector in this trial we developed and utilized a MR-guided gene delivery platform[20–24]. The primary aims of this trial were to demonstrate the safety of the procedure and detect biomarker evidence of increased brain AADC activity. Secondary aims were to evaluate clinical improvements in OGC and motor function after gene delivery.

The therapeutic approach described in this report has the potential to improve disease-related symptoms and motor function in patients with AADC deficiency. The success of this general strategy provides confirmation of targeted gene therapy as a tool to effectively treat other genetic disorders of the central nervous system (CNS). In particular, the techniques described in this study validate the use of efficient viral vectors, brain imaging, and

**Table 1 Baseline demographic and clinical characteristics of the 7 subjects.**

| Characteristic | Subject 1 | Subject 2 | Subject 3 | Subject 4 | Subject 5 | Subject 6 | Subject 7 |
|---|---|---|---|---|---|---|---|
| Age (y), sex | 9.0, F | 8.0, F | 5.9, M | 5.1, M | 6.7, F | 6.2, F | 4.5, M |
| Motor development | | | | | | | |
| Head control | No | Partial | No | No | Partial | No | No |
| Reach/grasp | No | No | No | No | Yes | No | No |
| Sit independently | No | No | No | No | No | No | No |
| Symptoms | | | | | | | |
| OGC | ++ | ++ | ++ | ++ | ++ | ++ | ++ |
| Irritability | ++ | + | + | ++ | + | + | + |
| Insomnia | ++ | ++ | ++ | + | - | ++ | + |
| *DDC* genotype[a] | c.714+4A > T; | c.714+4A > T; | c.714+4A > T; | c.19C > T p. (Arg7X); | c.714+4A > T; | c.714+4A > T; | c.782-3T > A; |
| | c.364C > T p.(Phe122Leu) | c.179T > C p.(Val60Ala) | c.1233dupT p.(Arg412Trp_fs) | c.304G > A p. (Gly102Ser) | c.1312T > C p.(Cys438Arg) | c.714+4A > T | c.1217T > A p.(Leu406Gln) |
| Follow-up duration | 38 months | 34 months | 30 months | 22 months | 20 months | 7 months | 7 months[b] |

*M* male, *F* female, *OGC* oculogyric crises, ++ severe, + mild or moderate, - absent.
[a]*DDC* reference sequence: NM_001082971.1(*DDC*).
[b]Subject 7 died 7 months after the procedure.

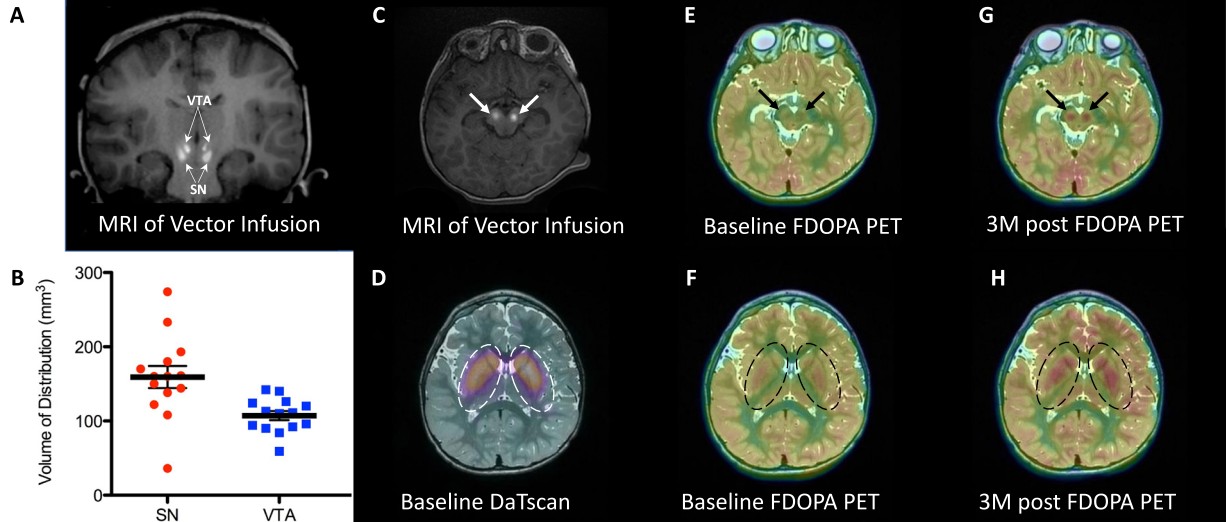

**Fig. 1 MR-guided delivery of AAV2-hAADC into the midbrain, baseline DaTscan and changes in FDOPA PET biomarker after gene delivery.** Coronal (**A**) and axial (**C**) MR images at the conclusion of the vector infusions into SN and VTA regions (white arrows). Bright signal corresponds to gadoteridol admixed with AAV2-hAADC. Infusions are performed sequentially while imaging, starting with right SN. Please, note accurate targeting and coverage in respective anatomical regions. **B** Coverage (volume of distribution, Vd) of all infusions performed into the SN and VTA in 7 subjects ($n = 2$ independent infusion sites (left and right) examined per participant ($n = 7$) for each target structure (SNc and VTA), for a total of $n = 14$ independent infusions per target structure). SN infusion (50 μL) achieved coverage of ~160 ± 60 mm³ (mean ± SD, $n = 14$ infusions (two infusions per participant)) with one suboptimal infusion due to leakage along the perivascular space. VTA infusion (30 μL) resulted in coverage of 103 ± 22 mm³. Coverage volume of gadoteridol (Vd) suggests almost 80% anatomical coverage of both SN and VTA in all subjects (except single SN in one patient). **D** DaTscan imaging of the striatum at baseline confirmed a normal pattern of dopaminergic innervation in all study subjects, indicative of preserved nigrostriatal pathway. **E**, **F** Baseline FDOPA imaging of the midbrain regions (SN and VTA, black arrows in **E**) and nigrostriatal projection (caudate nucleus and putamen, dotted line in **F**). Lack of signal in both regions represents impaired conversion of FDOPA to F-dopamine due to absent AADC activity. **G**, **H** Increased FDOPA PET uptake 3 months after AAV2-hAADC administration in the midbrain and striatum, respectively. Images for Subject 4 are shown as representative of the group; see Supplementary Fig. S1 for images for each individual subject. Source data are provided as a Source Data file.

optimized delivery tools as an overall framework for other clinical applications.

## Results

**Subjects.** Seven (7) subjects (4 female/3 male, age range 4–9 years, Table 1) were treated with viral gene therapy according to the study protocol (see Online Methods section). Subjects were enrolled sequentially into two dose groups: 3 subjects in Group 1 ($8.3 \times 10^{11}$ vg/mL, treated January–September 2017) followed by 4 subjects in Group 2 ($2.6 \times 10^{12}$ vg/mL, i.e. a 3-fold increase in vector concentration, treated May 2018–August 2019). The total infusate volume was 160 μL of vector for each subject. The duration of post-procedure follow-up is 24–36 months for the 3 subjects in Cohort 1, and 6–18 months for the 4 subjects in Cohort 2 (Table 1). The study was reviewed and approved by the Institutional Review Boards at the University of California San Francisco (Protocol No. 15-17756, approved on 24 June 2016) and The Ohio State University Wexner Medical Center (Protocol No. 2018H0269, approved on 29 November 2019).

**Target distribution of the vector.** Using real-time MR imaging, we were able to confirm accurate placement of the infusion catheter at each target (bilateral SNc and VTA) for all 7 subjects (Fig. 1A–C, see Online Methods for details). The pattern of distribution followed a roughly oval shape surrounding the catheter tip. In 1 subject, at one target, there was perivascular spread of the contrast agent along a brainstem vessel. Our previous studies have demonstrated that the distribution of the gadoteridol closely matches distribution of the infused AAV2 vector[19,22,23]. Based on anatomic boundaries identified on the pre-procedural MR scans, infusion of 50 μL into the SN resulted in mean coverage (mean ±

SD) of approximately 174 ± 49 mm³ with one suboptimal infusion due to leakage along the perivascular space which was excluded from analysis (n = 13). Infusion of 30 μL into VTA resulted in coverage of 107 ± 23 mm³ (n = 14), (Fig. 1B). The volume of distribution was 3 times the volume of infusion in the SN, and 3.4 times the volume of infusion in the VTA.

**Evidence of increased brain AADC activity after gene delivery.** We performed 6-[18F]-fluoro-L-DOPA (FDOPA) PET imaging at Baseline, Month 3, and Month 24 to look for evidence of a change in brain AADC activity after gene delivery. Integrity of the nigrostriatal pathway as described by Lee et al.[14] was confirmed by imaging of the dopamine transporter (DaTscan™) in all subjects at Screening (Fig. 1D). Pre-operative FDOPA PET demonstrated no evidence of FDOPA uptake in the striatum or other brain regions (Fig. 1E, F), consistent with absent AADC activity. Post-operative images at both Month 3 and Month 24 demonstrated increased FDOPA uptake in the midbrain (site of gene delivery, Fig. 1G, and Supplementary Fig. 1A) and also in brain regions that receive dopaminergic projections from the midbrain. A diffuse increase of FDOPA signal was detected in both putamen and caudate nucleus (striatum) (Fig. 1H and Supplementary Fig. 1B). Detection of increased FDOPA uptake in the striatum is consistent with our hypothesis that the AAV2-hAADC vector and AADC protein both undergo anterograde axonal transport via the intact nigrostriatal pathway into nigrostriatal terminals with accompanying enzymatic conversion of levodopa into dopamine.

Neurotransmitter metabolites were assayed in CSF on 2 separate occasions before the procedure (at screening and on the day of surgery), and four times after the procedure at months 3, 6, 12, and 24. (Fig. 2 and Table 2). At baseline, concentrations of the dopamine metabolite, homovanillic acid (HVA), were

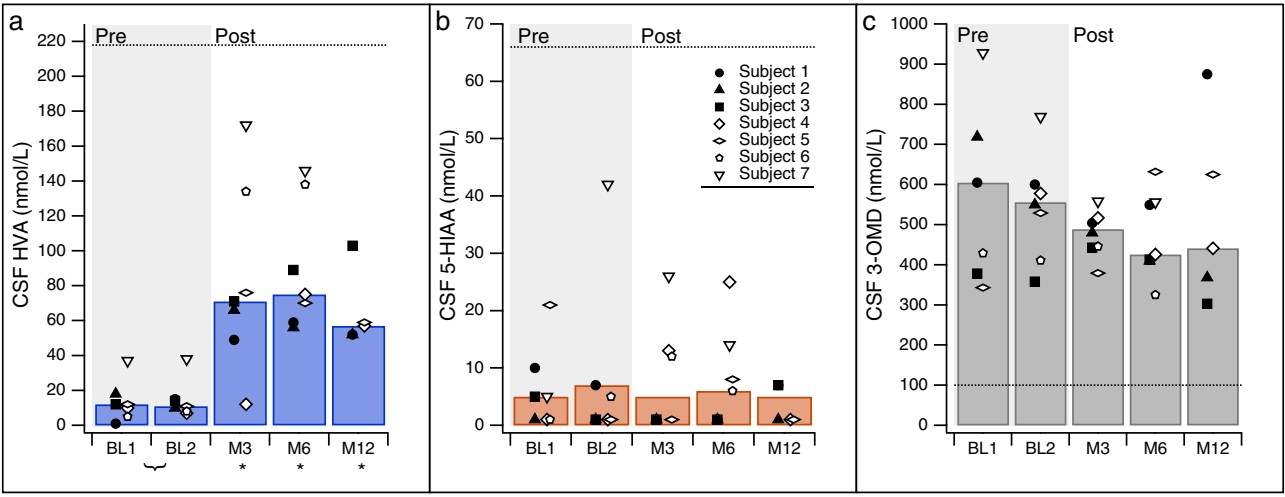

**Fig. 2 CSF Neurotransmitter metabolites.** Concentrations of CSF metabolites measured at 2 separate baseline (BL) timepoints, Month 3, Month 6, and Month 12, in individual subjects in Cohort 1 (low-dose, black markers); and Cohort 2 (high-dose, white markers) and summarized at each timepoint as the group median (bars). **A** Homovanillic acid (HVA), the dopamine metabolite, was significantly higher at each post-operative timepoint compared to the baseline mean (inverted bracket; (*$p = 0.0078$ at Months 3 and 6, $p = 0.0313$ at Month 12, one-tailed Wilcoxon signed-rank test). Lower limit of normal range: 218 nmol/L (dotted line). **B** 5-hydroxyindoleacetic acid (5-HIAA), the serotonin metabolite, did not change after gene delivery. Lower limit of normal range: 66 nmol/L (dotted line). **C** 3-O-methyldopa (3-OMD) was elevated in all subjects at all timepoints. Normal: <100 nmol/L (dotted line). Source data can be found in Table 2.

**Table 2 CSF neurotransmitter metabolites.**

| Metabolite (normal range) | Baseline | | Month 3 | Month 6 | Month 12 | Month 24 |
|---|---|---|---|---|---|---|
| | BL1 | BL2 | | | | |
| HVA (218–852 nmol/L) | | | | | | |
| Subject 1 | <5 | 15 | 49 | 59 | 52 | 12 |
| Subject 2 | 18 | 10 | 66 | 56 | 52 | 73 |
| Subject 3 | 12 | 14 | 71 | 89 | 103 | 137 |
| Subject 4 | 10[a] | 7 | 12 | 75 | 57 | – |
| Subject 5 | 12 | 11 | 76 | 70 | 59 | – |
| Subject 6 | 5 | 8 | 134 | 138 | – | – |
| Subject 7 | 37 | 38 | 172 | 146 | – | – |
| 5-HIAA (66–338 nmol/L) | | | | | | |
| Subject 1 | 10 | 7 | <5 | <5 | 7 | <5 |
| Subject 2 | <5 | <5 | <5 | <5 | <5 | 8 |
| Subject 3 | 5 | <5 | <5 | <5 | 7 | 24 |
| Subject 4 | <5 | <5 | 13 | 25 | <5 | – |
| Subject 5 | 21 | <5 | <5 | 8 | <5 | – |
| Subject 6 | <5 | 5 | 12 | 6 | – | – |
| Subject 7 | 5 | 42 | 26 | 14 | – | – |
| 3-OMD (<100 nmol/L) | | | | | | |
| Subject 1 | 605 | 600 | 504 | 549 | 875 | 443 |
| Subject 2 | 719 | 556 | 489 | 410 | 368 | 460 |
| Subject 3 | 378 | 358 | 443 | 413 | 303 | 432 |
| Subject 4 | 2104[a] | 578 | 517 | 426 | 441 | – |
| Subject 5 | 343 | 529 | 379 | 632 | 625 | – |
| Subject 6 | 429 | 411 | 446 | 325 | – | – |
| Subject 7 | 928 | 769 | 558 | 556 | – | – |

*HVA* homovanillic acid, *5-HIAA* 5-hydroxyindoleacetic acid, *3-OMD* 3-O-methyldopa.
[a]Subject 4 was taking carbidopa-levodopa at the time of the first baseline lumbar puncture. The medication was tapered off prior to the day of surgery (2nd baseline lumbar puncture).

<17% of the lower limit of normal in all subjects (Fig. 2a and Table 2). HVA concentration increased significantly 3 months after gene delivery compared to the mean of the two baseline measurements ($p = 0.0078$, one-tailed Wilcoxon signed-rank test, $n = 7$). This increase was sustained at Month 6 ($p = 0.0078$, $n = 7$) and Month 12 ($p = 0.0313$, $n = 5$) (Fig. 2a). CSF 5-hydroxyindolacetic acid (5-HIAA) concentration did not change after gene delivery (Fig. 2b and Table 2), consistent with expectations since serotonergic nuclei were not targeted in this procedure. 3-O-methyldopa (3-OMD), an alternate metabolite of levodopa, was elevated in all assays and did not consistently increase or decrease after gene delivery (Fig. 2c and Table 2).

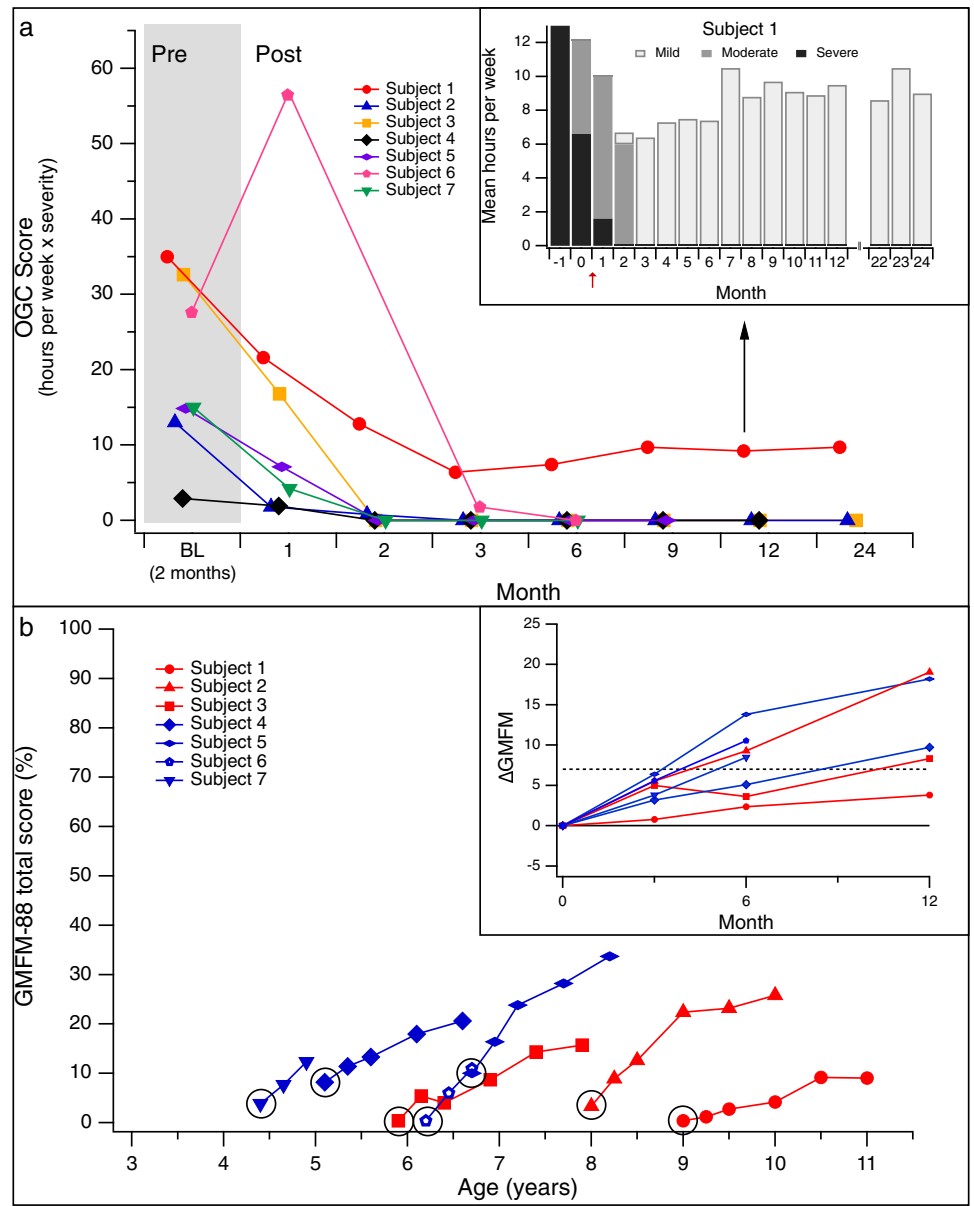

**Fig. 3 Changes in Oculogyric Crises (OGC) and motor function after gene delivery. A** The OGC score was calculated monthly, and represents the weekly average of the duration of episodes (hours per week), weighted by severity (grade 1–3: 1 = mild/eye deviation only; 2 = moderate/eye deviation + dystonia or dyskinesia of the face and/or neck; 3 = severe/dystonia or dyskinesia involving the trunk and/or limbs). By Month 3, OGCs completely resolved in 6/7 subjects. Inset (black arrow): Subject 1 had residual episodes throughout the 24-month study period; the severity decreased after surgery (red arrow). **B** Gross Motor Function Measure-88 (GMFM-88) scores for Cohorts 1 (red) and 2 (blue). Data are shown through Month 24 for Subjects 1–3, Month 18 for Subjects 4 and 5, and Month 6 for Subjects 6 and 7. Baseline GMFM scores for all subjects (circled) were ≤10, consistent with severe motor impairment. Inset: changes in GMFM score between Baseline and Month 12. An increase of ≥7 points (dotted line, representing a clinically meaningful positive change) was observed in 6/7 subjects by Month 12. Source data are provided as a Source Data file.

**Oculogyric crises improved in all subjects**. Subjects' caregivers recorded a log of the duration and severity of each OGC observed from 2 months before surgery through the 24-month follow-up period. We summarized the average burden of OGC for each month, using a score that represents the weekly duration (in hours) weighted by the severity (grade 1–3) of episodes. For example, 5 h per week of moderate episodes (grade 2: eye deviation associated with involuntary facial movements) would be represented by a score of 10, while 10 h per week of severe symptoms (grade 3: whole body dystonia) would result in a score of 30. All subjects experienced several hours per week of OGC at baseline (OGC score range: 3.2–35).

After gene delivery, OGC ceased completely in 6/7 subjects. Resolution occurred within 9–33 days of surgery for 5 subjects, and within 90 days for 1 subject (Fig. 3a). This resolution was sustained throughout the study follow-up period. In all, 1 subject had residual episodes of significantly reduced severity compared to baseline (Fig. 3a, insert) throughout 24 months of post-surgery follow-up. The marked improvement in OGC provided clear clinical evidence of increased brain AADC activity that began as soon as 9 days after gene delivery.

**Motor function and developmental milestones**. Gross motor function was evaluated using the Gross Motor Function Measure

**Table 3 Developmental milestones after gene delivery.**

| Subject | Follow-up duration (months) | Head control | | Sit independently | | Reach and grasp | | Walk with 2-hand support | |
|---|---|---|---|---|---|---|---|---|---|
| | | Baseline | Post-surgery (month attained) | Baseline | Post-surgery (month attained) | Baseline | Post-surgery (month attained) | Baseline | Post-surgery (month attained) |
| 1 | 36 | No | No | No | No | No | No | No | No |
| 2 | 33 | ± | Yes (2) | No | Yes (7) | No | Yes (10) | No | Yes (17) |
| 3 | 30 | No | Yes (12) | No | Yes (21) | No | No | No | No |
| 4 | 22 | No | Yes (7) | No | Yes (11) | No | Yes (10) | No | No |
| 5 | 20 | ± | Yes (3) | No | Yes (7) | Yes | Yes | No | Yes (18) |
| 6 | 7 | No | Yes (2.5) | No | Yes (7) | No | No | No | No |
| 7 | 7 | No | Yes (4) | No | – | No | – | No | – |

(GMFM-88), a standardized instrument designed to assess changes in motor function over time in children with motor impairment due to cerebral palsy[25]. The total score, which ranges from 0 to 100 percentage points, is derived as an unweighted average of scores in five dimensions (lying and rolling; sitting; crawling and kneeling; standing; and walking, running, and jumping). The GMFM-88 was selected for this study as a tool to assess motor function because it was developed to assess children with motor impairment due to an insult to the developing brain early in life (cerebral palsy), as occurs in AADC deficiency, and because the GMFM can accurately test and measure changes in children who have severe motor impairment.

The baseline GMFM-88 total scores of the 7 subjects ranged from 0.4 to 10 points, consistent with severe impairments in gross motor function (Fig. 3b). All subjects achieved recognizable gains in motor function after the procedure, manifested by increased tone and improvements in head and trunk control and purposeful limb movements. The rate of improvement varied considerably from 1 subject to another. An increase in GMFM score of at least 7 points, which is considered to be a clinically meaningful improvement[25], was observed in 4/7 subjects (57%) at Month 6, and 6/7 subjects (86%) by Month 12 (median 9.7, range 3.8–19.1 points; Fig. 3b, inset). All 5 subjects (100%) who had at least 18 months of post-surgical follow-up (Subjects 1–5) had a >7-point increase in GMFM-88 score at Month 18 (median 13.9, range 8.6–23.7 points).

Motor function improvements after gene delivery were accompanied by the attainment of developmental milestones. Head control was attained by 4/7 subjects (57%) by Month 6, and 6/7 subjects (86%) by Month 12. Independent sitting was attained by 4/7 subjects (57%) by Month 12 (Table 3 and Supplementary Movies 1–3). At 12 months, 3/7 subjects (43%) could reach and grasp (acquired after surgery in 2 of 3). Subjects 2 and 5 (baseline ages 8 and 6 years, respectively) were additionally able to walk with trunk support by Month 12, and to walk with 2-hand support by Month 18. Two and a half (2.5) years after surgery, Subject 2 started to take independent steps (Supplementary Movie 1).

By Month 12, Subjects 2 and 5 were able to eat and drink by mouth, and Subject 2 gained the ability to speak ~50 single words. Subject 5 learned to communicate using an augmentative communication device between 12 and 18 months after gene delivery (Supplementary Movie 2). These functional gains were reflected in raw score increases at 12 months across all domains (Communication, Daily Living Skills, Socialization, and Motor) of the Vineland Adaptive Behavior Scales 2nd edition for Subjects 2 and 5 (Supplementary Table 1).

**Improvement in non-motor symptoms of AADC deficiency.** A standard checklist of symptoms of AADC deficiency was

evaluated by study investigators, via parent interview, at baseline and at each follow-up evaluation. Each symptom was rated as 'major' (frequent and/or severe), 'minor' (infrequent and/or mild), or absent

Prior to gene delivery, all patients had a mood disturbance characterized by frequent crying, irritability, or agitation. In 3/7 subjects the mood symptoms were severe (Fig. 4a). Caregivers periodically completed a 7-day sleep and behavior diary that documented a transient increase in irritability for some subjects 1–3 months after surgery, followed by marked sustained improvement in mood in all subjects (Fig. 4a, b and Supplementary Movie 3).

Five (5) of 7 subjects presented with severe sleep disturbance at baseline, and parents subjectively reported an improvement in their child's sleep after surgery (Fig. 4c). It was difficult to quantify this improvement based on data recorded in the caregiver diary for most cases, but an overall trend towards an increase in the number of hours of sleep per night can be observed, and for Subject 2, the improvement in sleep quantity was dramatic (Fig. 4d).

After gene delivery, parents also reported marked improvement in sweating, feeding difficulties (such as vomiting), and upper airway obstruction due to profuse oral secretions, nasal congestion, and stridor (Fig. 4e–g). The improvement in feeding tolerance in Subject 4 led to an increase in body weight from four standard deviations below the mean for age at Baseline, to the 2nd percentile at 22 months post-surgery. All subjects were receiving enteral nutrition via gastrostomy at Baseline. Body weight at Baseline for the other subjects ranged from the 6th–32nd percentile, and remained stable over the course of follow-up.

**Safety outcomes.** All 7 subjects tolerated the surgical procedure without any direct short-term or long-term adverse effects. In all, 1 subject was noted to have a focal depression of the calvarium at a cranial pin site on a post-operative imaging study but this did not affect the brain parenchyma and did not result in any other consequences. There were no intracranial hemorrhages, strokes, or infections related to the procedures. There were no recorded serious adverse events related to the study intervention (Table 4).

Beginning 3–4 weeks after gene delivery, all subjects experienced a transient worsening of irritability and sleep dysfunction which improved within weeks and eventually reached a state that was better than baseline (Fig. 4b, d). At 3–4 weeks, all subjects also developed anticipated involuntary movements, or dyskinesia. These motor and behavioral symptoms coincided with the initial AAV-driven expression of AADC and subsequent abrupt increase in dopamine synthesis. Dyskinesia peaked in severity between 1 and 3 months after surgery, and gradually improved over subsequent months (Fig. 5). As dyskinesia emerged, dopaminergic and anticholinergic medications were tapered in 6/7 subjects

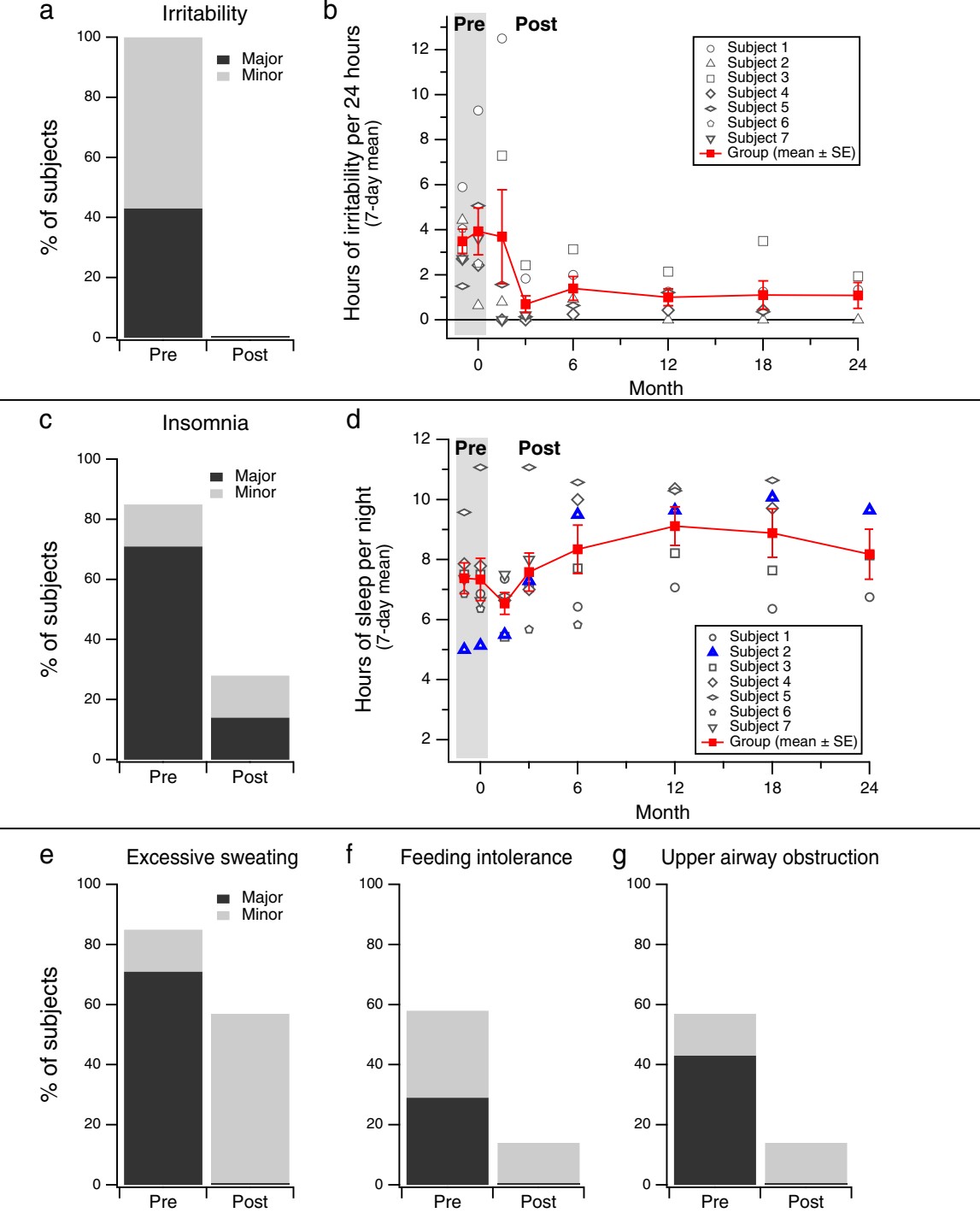

**Fig. 4 Non-motor symptoms before and after gene delivery. A** The percentage of subjects ($n = 7$) who, by caregiver report, experienced irritability at baseline compared to the last available follow-up timepoint (Month 24 for Subjects 1–3, Month 18 for Subjects 4 and 5, and Month 6 for Subjects 6 and 7). **B** The number of hours or irritability per day recorded in caregiver diaries increased slightly immediately after surgery, and then improved on average (mean ± SE). **C** Prevalence and severity of insomnia before and after gene delivery by caregiver report. **D** The mean number of hours (±SE) of night-time sleep recorded in caregiver diaries improved slightly across subjects after gene delivery. For Subject 2 (blue triangles), the improvement was dramatic. **E**–**G** Prevalence and severity of other non-motor symptoms before and after gene delivery by caregiver report (see details in **A**, above). Major (black): frequent and/or severe, with significant impact on comfort or function; Minor (gray): infrequent and/or mild. Source data are provided as a Source Data file.

**Table 4 Summary of adverse events and serious adverse events.**

|  | Adverse events[a] | Serious adverse events[b] | Fatality |
|---|---|---|---|
| Related to gene therapy |  |  |  |
| Skin irritation on cheeks due to dyskinesia | 1 | – | – |
| Related to study procedures |  |  |  |
| Vomiting following administration of potassium iodide prior to DaTscan | 1 | – | – |
| Post-operative anemia requiring transfusion | 2 | – | – |
| Related to AADC deficiency |  |  |  |
| Sudden death of unknown cause | – | – | 1 |
| Hypoglycemia following sedation for study procedure | 2 | 1 | – |
| Non-specific |  |  |  |
| Viral respiratory tract infection | – | 5 | – |
| Hematemesis associated with gastro-esophageal reflux | – | 1 | – |
| Pneumonia | – | 2 | – |
| Urinary tract infection | – | 1 | – |
| Skin rash | 2 | – | – |
| Eye infection/irritation | 2 | – | – |
| Gastroenteritis | 1 | – | – |
| Vomiting due to missed doses of anti-reflux medication | 1 | – | – |
| Total | 12 | 10 | 1 |

AADC aromatic L-amino acid decarboxylase deficiency.
AADC is a rare neurodevelopmental disorder. Here the authors describe a clinical trial of MR-guided delivery of AAV2-AADC for the treatment of AADC.
[a]Adverse events: classified as mild or moderate.
[b]Serious adverse events: classified as severe or medically significant, requiring or prolonging hospitalization.
Aromatic L-amino acid decarboxylase deficiency (AADC) is a rare neurodevelopmental disorder. Here the authors describe a clinical trial of MR-guided delivery of AAV2-AADC for the treatment of AADC.

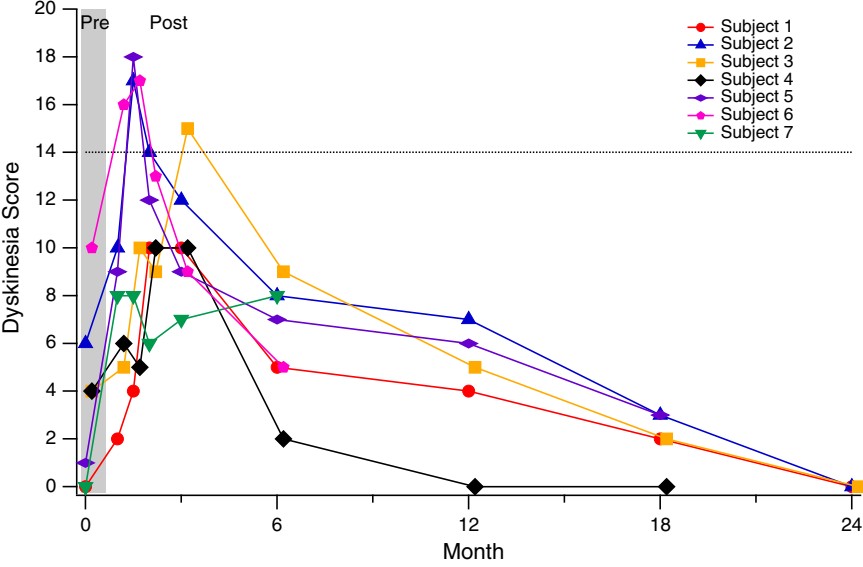

**Fig. 5 Time course of dyskinesia after gene transfer.** Dyskinesia score, adapted from the Abnormal Involuntary Movement Scale (AIMS). Involuntary movements in each of 7 body regions were scored from 0 (none) to 4 (severe), for a maximum possible score of 28. A total score of 14 (dotted line) may correspond with either mild generalized dyskinesia, or moderate-severe involuntary movements in a more focal distribution. Dyskinesia peaked in severity between 1 and 3 months after surgery, and gradually improved thereafter. At Month 24, dyskinesia had resolved completely in all 3 subjects in Cohort 1.

and 3/7 stopped all dopaminergic medications. No additional medications were prescribed to treat the involuntary movements. At Month 24, none of the 3 subjects in Cohort 1 had any residual dyskinesia (Fig. 5).

One subject (Subject 7) died suddenly and unexpectedly at home without any preceding illness, 7 months after surgery. The subject had completed all study evaluations through Month 6, when he appeared to be in good health, and demonstrated improvements in symptoms and motor function. The cause of death was unknown, and was judged to be most likely attributable to the underlying primary disease.

## Discussion

MR-guided convection-enhanced delivery (CED) of AAV2-hAADC into the midbrain can be performed safely in children with AADC deficiency. Children with AADC deficiency experience a combination of disabling motor and autonomic symptoms, and severe motor and developmental disability. Minimal spontaneous improvement occurred in our study subjects prior to enrollment. All 7 subjects exhibited measurable clinical improvements in both symptoms and motor function following gene delivery. OGC resolved completely in 6 of 7 subjects. Although there was some variation in the rate and degree of

motor function improvement, 4 of the 5 subjects who had been followed for at least 18 months after gene delivery gained the ability to sit independently, and 2 of the 5 gained the ability to walk with 2-hand support within that time period. Substantiating these clinical observations, there was associated biomarker evidence of increased brain AADC activity by measurement of HVA in CSF and FDOPA PET imaging.

Direct infusion of AAV2-hAADC into the SNc and VTA is technically feasible and was performed without surgical complications in this series of patients. Successful delivery of the *DDC* gene therapy vector required identification of the anatomic targets, accurate placement of an intracranial catheter, and subsequent confirmation of vector delivery. This approach was accomplished through the use of real-time MR imaging in conjunction with co-infusion of a MR contrast agent. Pre-operative identification of the SNc and VTA using high-resolution MR imaging allowed for planning of catheter trajectories designed to prevent passage through eloquent regions. An MR-guided navigation system allowed direct intra-operative confirmation of catheter placement into the selected targets. Finally, vector infusion was monitored with continuous MR imaging that confirmed accurate and reproducible distribution of the infusate.

Infusion of AAV2-hAADC resulted in measurable changes in clinical function and dopamine production suggesting that restoration of AADC activity in physiological regions (midbrain) can reverse the pathologic phenotype observed in primary AADC deficiency. We observed dramatic improvements in symptoms, including OGC, irritable mood, and sleep dysfunction. These changes became apparent soon after gene delivery, typically within the first 3 months. Caregivers reported that these changes, particularly the resolution of OGC and improvements in sleep, had a major positive impact on the child's and family's quality of life.

Improvements in motor function unfolded over a longer time course, and the rate of improvement varied considerably between subjects. Greater improvement was not obviously associated with either higher vector dose (Cohort 2) or with younger age in this small group of subjects. All 7 subjects had severe motor impairment at baseline, with severe hypotonia, minimal voluntary movements, and lack of head control in most cases. After gene delivery, the time to attain head control in 6 subjects ranged from 2 to 14 months, while the time to attain independent sitting ranged from 7 to 21 months. GMFM-88 total scores improved by more than 7 points in all subjects, and in 2 subjects (baseline ages 6 and 8 years) who gained the ability to walk with 2-hand support, eat and drink by mouth, and communicate either verbally or with the aid of an augmentative communication device, the observed change in score was >20 points. This magnitude of motor function improvement in our subjects would not be expected to occur in the natural course of AADC deficiency for patients in this age range (4–9 years) who have severe motor impairment. In a retrospective study of 63 patients (age range 6 months to 36 years), 11 subjects with a mild-moderate disease phenotype were able to walk independently[4]. In those 11 subjects, the potential for motor development was evident in early childhood, with all having attained the ability to sit independently by age 4 years. In contrast, the subjects in our gene therapy trial had absent head control at Baseline ages of 4 years and above, and therefore would not have been expected to proceed to sit, stand, or walk independently based upon what is known about the typical course of developmental outcomes in this disease.

We propose that the broad range of clinical benefits we observed are attributable to physiological restoration of dopamine synthesis, storage, release, and reuptake in the pre-synaptic terminal projections arising from midbrain dopaminergic neurons. Dopaminergic neurons in SNc and VTA send highly arborized projections into the striatum and nucleus accumbens[26]. Thus, AAV2-AADC transduction of SNc and VTA neurons would be expected to result in restoration of AADC activity in widespread regions outside the midbrain, as demonstrated by the distribution of changes in FDOPA PET uptake that we observed in the caudate nucleus and the putamen in our subjects after gene delivery.

Two groups in Taiwan and Japan have previously investigated infusion of AAV2-AADC into the bilateral putamen[15–17]. Concentrations of CSF HVA after surgery remained below 25% of the normal range in the majority of subjects[15,16]. OGC were reported to become less severe and less frequent after dosing. In the cohort of 10 patients described by Chien et al, who were on average younger than the subjects in our cohort, 6/10 subjects had gained head control at 12 months, and 2/8 could sit independently after 2 years[16]. In the 6-subject cohort described by Kojima et al.[17], which included 4 subjects over age 10 years, none gained the ability to sit independently at 2 years.

Diffuse restoration of AADC activity in dopamine terminals in both the caudate nucleus and pre- and post-commissural putamen, as seen after midbrain administration of AAV2-hAADC in our study, is likely to underlie the consistent improvement in both motor and non-motor symptoms that we observed in our subjects. We suspect that dopamine transmission in mesocortical and mesolimbic pathways may also have improved, but detection of AADC activity in anatomical regions such as the nucleus accumbens and amygdala was unfortunately constrained by the limits of PET resolution possible in this study.

HVA levels in the CSF constitute a direct biomarker of dopamine metabolism. Low baseline levels of HVA represent deficient activity of endogenous AADC and point to failure of dopamine synthesis. Restoration of AADC in midbrain dopaminergic neurons leads to significant and consistent increases in HVA concentration. We believe that restoration of this dopaminergic biomarker in the CSF is directly correlated with diffuse and physiological activity of AADC in the dopaminergic brain circuitry in AADC deficient children. We did observe a persistent elevation of 3-OMD after gene delivery in all subjects, which is consistent with some degree of residual AADC deficiency. The marked clinical improvements we observed in our subjects therefore did not require complete restoration of AADC function, but the localization of that function, not only the extent, may be a key determinant of clinical outcomes. Serotonergic brain circuitry remained unaffected as demonstrated by lack of 5-HIAA increase in the CSF since we restricted our gene delivery to the midbrain, without any targeting of brain stem nuclei.

The durability of AADC transduction for children with AADC deficiency is unclear, but clinical data from subjects with PD (>4 years) and pre-clinical data from non-human primates (>8 years)[27–30] suggest that long-term and possibly life-long expression persists.

Gene therapy, i.e. gene replacement with functional genes, is a potentially revolutionary therapeutic approach for many human diseases. However, many technical hurdles exist for treatment of complex diseases which include optimization of gene expression, anatomic localization of gene expression, and availability of appropriate endpoints to measure efficacy. From a technical perspective, AADC deficiency has several favorable features. First, gene replacement and expression in a specific small anatomic area, midbrain dopaminergic neurons, has a large clinical impact. This gene therapy clinical study is the first designed to prospectively utilize axonal transport of viral vectors to disseminate the therapeutic gene along brain circuitry affected by the gene mutation. Second, the surgical and imaging tools currently available, particularly MR-guided CED, permit highly accurate delivery to virtually any region of the CNS with a favorable risk

profile. And finally, direct imaging by FDOPA PET provides a non-invasive tool to assess enzyme activity in the treated subjects. Although this was an early phase clinical trial with safety as the primary objective, the results obtained are consistent with successful *DDC* gene delivery and expression in the human CNS. The gene therapy approach described here represents many years of careful work to develop and to understand vector tropism, axonal transport, methods, and devices for gene delivery and application of MRI technology to maximize safety and gene delivery. This work provides a framework for the treatment of other human CNS genetic diseases, and iterative refinement of the individual components of this approach will facilitate broader application.

## Methods

**Study design**. Our study complies with all ICMJE guidelines for reporting clinical trials. This study was an adaptive, single-stage dose-escalation, open-label safety, and efficacy study of AAV2-hAADC for the treatment of children with AADC deficiency (ClinicalTrials.gov. identifier NCT02852213). The primary aims were to demonstrate the safety of the procedure and document biomarker evidence of restoration of brain AADC activity. Secondary aims were to assess clinical improvement in symptoms and motor function.

Subjects were enrolled sequentially into two dose groups: 3 subjects in Cohort 1 ($8.3 \times 10^{11}$ vg/mL) followed by 4 subjects in Cohort 2 ($2.6 \times 10^{12}$ vg/mL). The total infusate volume was 160 μL of vector for each subject. The vector concentration was 3x higher for subjects in Cohort 2. We lowered the minimum age for inclusion in the trial from 5 years to 4 years for Cohort 2 after demonstrating the safety of the surgical procedure in Cohort 1. The minimum age criterion was in place to ensure adequate skull maturity to tolerate the head fixation procedure required during surgery.

Screening, treatment, and evaluation procedures were performed at UCSF Benioff Children's Hospital and The Ohio State University Wexner Medical Center. Due to the ultra-rare nature of AADC deficiency a sample size of 7 subjects was selected. Seven (7) subjects were treated with viral gene therapy according to the study protocol. Subject 6 was the 6th subject to be screened for the study, but was the final one (after Subject 7) to undergo surgery due to 2 severe respiratory infections that delayed the scheduling of surgery following the initial screening visit.

Written informed consent was obtained from the legally authorized representative of all study participants. The study was reviewed and approved by the Institutional Review Boards at the University of California San Francisco (Protocol No. 15-17756, approved on 24 June 2016) and The Ohio State University Wexner Medical Center (Protocol No. 2018H0269, approved on 29 November 2019). Videos containing identifying information are published with the consent of each participant's legally authorized representative.

**Participants**. Male and female subjects aged 4–9 years with a confirmed diagnosis of AADC deficiency who met all inclusion/exclusion criteria were enrolled (Enrollment dates - First participant: 2 December 2016; Last participant: 15 October 2018). Subjects were eligible if they had severe motor impairment (defined as inability to walk, with or without an assistive device), despite treatment with standard medical therapy for AADC deficiency. The main exclusion criteria were medical co-morbidities judged to confer excessive surgical risk (such as baseline requirement for home ventilatory support, structural brain malformation, and coagulopathy), a history of previous stereotactic neurosurgery, and receipt of any investigational agent within 60 days prior to Baseline and during study participation.

**Viral vector**. The AAV2-hAADC vector consists of an adeno-associated virus, serotype 2 (AAV2) containing human AADC complementary DNA (cDNA), human cytomegalovirus (CMV) promoter, and 3′UTR sequences. AAV2-hAADC was supplied in 0.5 mL aliquots as a suspension at a stock concentration of $4.9 \times 10^{12}$ vector genomes per mL. The vector and excipient for dilution were supplied to the clinical site by the Clinical Vector Core, Children's Hospital of Philadelphia.

**Surgical procedure**. The surgical procedure was performed in a 3T interventional MRI suite. A combination of volumetric T1, slab T2, and slab inversion recovery sequences were acquired prior to the procedure and bilateral VTA and SNc infusion sites and trajectories were identified using standard software packages (iPlan v3.0, Brainlab, Germany). The trajectories were designed to avoid cortical veins and arteries, and the ventricular system. Following general anesthesia, the subjects' heads were immobilized with an MRI-compatible head-holder. A bicoronal scalp incision was made and two small craniotomies created in the right and left frontal regions using the pre-selected entry points. The patient was then transported to the intra-operative MR unit and skull-mounted aiming devices (SmartFrame®, ClearPoint Neuro Inc. (formerly MRI Interventions Inc.)) were mounted over each craniotomy site. The trajectory and depth were refined based

on intra-operative MR images acquired after placement of the targeting device. An end-lumen ceramic and fused silica cannula with a progressive bullet-shaped step design (ClearPoint Neuro Inc. (formerly MRI Interventions Inc.)) connected to an infusion pump with a syringe was inserted into the SNc. The location of the catheter was confirmed with a high-resolution MR scan prior to vector infusion. Fifty microliters (50 μL) of the AAV2-hAADC vector and 2 mM of a chelated gadolinium contrast agent (Gadoteridol (ProHance®, Bracco)) were infused into the SNc by convection-enhanced delivery (CED), or pressurized infusion. The catheter was withdrawn and repositioned using new target coordinates for the VTA. This site was infused with 30 μL. A similar infusion protocol was repeated on the opposite side. The total infusate volume was 160 μL of vector for each subject (80 μL per hemisphere). Calculations were made of the total volume of infusate and percentage of VTA and SN coverage by gadoteridol signal.

**Outcome measures**. Primary outcome measures assessed safety and biomarker evidence of increased brain AADC activity. Safety of the procedure was evaluated by brain MRI 48 h post-surgery, caregiver report of symptoms at each study visit, neurologist rating of post-surgery involuntary movements (dyskinesia) at each study visit, and caregiver diary of sleep and behavior symptoms at selected visits. Evidence of biological AADC activity was measured by [$^{18}$F]FDOPA PET and analysis of CSF neurotransmitter metabolites before and after surgery. Secondary outcome measures assessed clinical efficacy as expressed by improvements in symptoms and motor function. Caregivers kept a detailed log of the duration and severity of all OGC (OGC Log) before and after surgery. A checklist of AADC deficiency-related symptoms was reviewed at each study visit. Caregivers periodically recorded sleep and behavior observations in a 7-day diary. Gross motor function was evaluated using the Gross Motor Function Measure-88 (GMFM-88). All measures are described in further detail below, grouped by category (clinical/laboratory/imaging).

*Clinical measures*. A systematic assessment of adverse events and side effects was performed at each visit with the study neurologist (Screening, Baseline, Weeks 1, 2, 4, 5, 6, 7, 8, and Months 3, 6, 12, 18, and 24). Information was recorded on an Adverse Events (AE) Log which included type of AE, dates of onset and resolution, severity, and perceived relationship to experimental therapy (Table 4). The severity of each AE was rated based on the NCI Common Terminology Criteria for Adverse Events v4.0 (CTCAE; https://ctep.cancer.gov/protocolDevelopment/electronic_applications/docs/CTCAE_4.03.xlsx). A general medical and a full neurologic examination was conducted at each study visit with selected examinations recorded on video.

At each study visit with the neurologist (see timepoints above), a standard checklist of AADC deficiency-related symptoms was reviewed, in which parents were asked to judge the severity of each symptom as "major" (frequent and/or severe), "minor" (infrequent and/or mild), or absent. Subjects were also examined by the neurologist at each visit to assess the severity of involuntary movements, or dyskinesia. A dyskinesia score, based on the Abnormal Involuntary Movement Scale (AIMS) was calculated by assessing the severity of involuntary movements in each of 7 body regions on a scale of 0 (none) to 4 (severe), for a maximum possible score of 28 (Fig. 5).

Caregivers documented the duration and severity of all OGC that occurred during the study period, beginning 2 months prior to surgery, in an OGC Log. OGCs typically occur 2–3 times per week and each episode lasts from 30 min to 8 h; therefore, the average weekly duration and severity of episodes can be accurately and reliably quantified. We created a 3-point scale to describe episode severity based on clinical experience: Grade 1 (mild) = eye deviation only; Grade 2 (moderate) = eye deviation associated with involuntary facial movements; Grade 3 (severe) = eye deviation associated with dyskinesia or dystonia of the trunk and/or limbs. Parents were trained at the initial screening visit to complete the OGC log by documenting a start time, end time, and severity score (1–3) for each episode. They then prospectively recorded all observed episodes throughout the study period. An OGC severity score, consisting of a product of the duration (hours) and severity (grade 1–3) was calculated by summing the number of hours of each severity category per calendar month, and then calculating the weekly (7-day) average score for that month. For example, a score of 15 may represent 5 h per week of severe (severity score 3) symptoms, 15 h per week of mild (severity score 1) symptoms, or an intermediate number of hours of symptoms of mixed severity.

A standardized behavior and sleep diary was used to capture sleep and mood dysfunction resulting from brain dopamine dysregulation. Subject's caregivers were trained in the completion of a symptom diary by study center personnel. The symptom diary was completed twice during the period between the Screening and Baseline visits, each over a consecutive 7-day period at least 1 month apart. Follow-up observations for 7-day periods were recorded at Week 6 and Months 3, 6, 12, 18, and 24 after surgery.

Gross motor function was evaluated using the GMFM-88. The total score was derived as an unweighted average of the five dimension scores: lying and rolling (17 items); sitting (20 items); crawling and kneeling (14 items); standing (13 items) and walking, running, and jumping (24 items). Each dimension score was defined as percentage of maximum score for the dimension in question, and the maximum possible total score was 100%. The evaluations were performed by a pediatric physical therapist and recorded on video at Baseline, 3, 6 12, 18, and 24 months after gene delivery. The assessments were then independently scored by two

therapists, the one who had performed the test in-person, and a second therapist who reviewed each recorded session on video (post-hoc, blinded to the timepoint of each assessment). When the total scores of the two independent assessments were within 2% points of each other, the score of the original (in-person) assessor was retained. When there was a difference of >2 points between the total scores, a third reviewer (TSP) reviewed the video to determine each individual item's final score and ensure accurate and consistent scoring across subjects.

The parents of all the patients participated in a formal survey interview using the Vineland Adaptive Behavior Scale, Second Edition (VABS-II), a measure of adaptive behavior and developmental functioning for individuals from birth through 90 years of age[31]. This tool is administered to the primary caregiver of an individual being assessed and provides information in various domains of functioning. Specifically, there are four domains (Communication, Daily Living Skills, Socialization, and Motor Skills), up to 11 subscales (three each for all domains except Motor, which has two), and an overall Adaptive Behavior Composite Score. Standard scores (population mean = 100, SD = 15) are produced for the domain and composite scores, scaled scores (called v-scale scores, with a population mean = 15, SD = 3) are produced for the subscale scores, age equivalents can also be generated and may be combined to determine an overall Adaptive Behavior Composite score. Strong internal consistency, test–retest reliability, and validity have consistently been demonstrated. However, due to the significant global developmental delays and limited abilities of our subject cohort, there was no utility in calculating formal standard, scaled, and/or composite scores. Instead, raw score comparisons were reviewed across time points to assess any positive developmental progress (Supplementary Table 1). The VABS-II scale was administered at Baseline, 12 and 24 months after surgery.

*Laboratory analysis*. CSF was collected by lumbar puncture with HPLC analysis of the following metabolites: homovanillic acid (HVA), 5-hydroxyindoleacetic acid (5-HIAA), 3-O-methyldopa (3-OMD), and 5-methyltetrahydrofolate (5-MTHF). Plasma AADC enzyme activity was determined by quantitative HPLC method[32,33]. Both analyses were performed by MNG Laboratories, a LabCorp Company (Atlanta, GA).

*Imaging studies*. MR scans were performed pre-operatively as part of the screening process, during the infusion procedure, and then at 48 h following the surgical procedure, with a final scan obtained 24 months following the procedure. The 48-h MR scan used a 'rapid' protocol and consisted of a 3D T2 FSE sequence, and axial diffusion.

6-[$^{18}$F]-fluoro-L-DOPA (FDOPA) is a PET radiopharmaceutical used to image dopamine synthesis capacity. PET scans were performed at screening, and at 3 months, with a final study planned 24 months after the procedure. The pediatric dose of 0.3 mCi for FDOPA was based on 10% of the adult dose of 3 mCi (UCSF RUA approval: RU133031, OSU approval: 2018H0269). PET scans were analyzed for quantification of AADC activity and differences in the distribution of AADC activity between baseline and post-surgery scans.

[$^{123}$I]ioflupane (DaTscan™) selectively binds to pre-synaptic dopamine transporters and provides a method for imaging nigrostriatal terminals in the striatum. DaTscan™ is a FDA-approved radiopharmaceutical used in conjunction with single photon emission computed tomography (SPECT) scan for use in adults. A DaTscan was performed at the screening visit to document nigrostriatal pathway integrity.

*Statistical analysis*. Increases in CSF HVA from Baseline to Month 6 and Baseline to Month 12 were analyzed using a one-tailed Wilcoxon signed-rank test.

*Reporting summary*. Further information on research design is available in the Nature Research Reporting Summary linked to this article.

## Data availability

De-identified individual participant data that underlie the results reported in this article as well as the study protocol and analytic code will be shared beginning 3 months and ending 5 years following article publication to researchers who provide a methodologically sound proposal and only to achieve aims in the approved proposal. Proposals should be directed to Krzysztof.Bankiewicz@osumc.edu. To gain access, data requestors will need to sign a data access agreement. Source data are provided with this paper.

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

## Acknowledgements

This study was supported by the NINDS/NIH (R01NS094292), NIH/NINDS BrIDGs Program (1 × 01 NS073514-01), the AADC Research Trust, the Pediatric Neurotransmitter Disease Association to K.S.B., and funding from The Ohio State University. We would like to thank the Advisory Committee (Darryl De Vivo, Ronald Crystal, Roser Pons, Manju Kurian, Un Jung Kang, Bernard Ravina, and Jonathan Mink) for useful comments regarding conduct of this clinical trial; Bradley Schlaggar for critical review of the manuscript; Diane Balderson and Susan Walker for assistance with data interpretation and statistical analysis; Robert Palisano for consultation and training on the GMFM-88; Todd Dubnicoff for assistance with video editing; and ClearPoint Neuro Inc. for generous donations of stereotaxic devices, infusion cannulas, and technical support.

## Author contributions

K.S.B. (Sponsor–Investigator), N.G., T.S.P., and W.S.S. participated in design of the study, data collection, oversight of the clinical trial, data analysis, data interpretation, and drafted the manuscript. N.G., J.B.E., R.L., K.S.B., J.L., and P.L. performed the neurosurgical procedure. T.S.P., A.V., A.G.P., and A.F. performed neurological evaluations. N.S., E.S., G.O.B., and J.C.H. conducted and scored motor function evaluations, and S.M.L. conducted neuropsychological evaluations. J.I.C., W.S.S., and A.M. provided study coordination, Y.S. and M.P. performed neuroimaging evaluation, and K.H. performed C.S.F. neurochemistry evaluations. All authors reviewed and approved the manuscript prior to submission.

## Competing interests

N.G. reports relationships with Oscine Therapeutics (consulting) and Y-mAbs Therapeutics (consulting). K.H. reports that he is employed by Medical Neurogenetics Laboratories, a company that provides commercial diagnostic testing for aromatic L-amino acid decarboxylase deficiency. P.L. reports relationships with Axovant (Advisory Board), Neurocrine Biosciences (research funding, consulting), UniQure (research funding), Voyager Therapeutics (research funding), and Clearpoint Neuro (consulting). K.S.B. is the founder and equity holder of Brain Neurotherapy Bio. The remaining authors have no competing interests to disclose.

## Additional information

[1]Department of Neurological Surgery, University of California San Francisco, San Francisco, CA, USA. [2]Department of Neurology, Washington University School of Medicine, St. Louis, MO, USA. [3]Department of Neurology, University of California San Francisco, San Francisco, CA, USA. [4]Department of Rehabilitative Services, University of California San Francisco, San Francisco, CA, USA. [5]Department of Pediatrics, University of California San Francisco, San Francisco, CA, USA. [6]Department of Radiology and Biomedical Imaging, University of California San Francisco, San Francisco, CA, USA. [7]Medical Neurogenetics Laboratories, Atlanta, GA, USA. [8]Therapy Services, St. Louis Children's Hospital, St. Louis, MO, USA. [9]School of Health and Rehabilitation Sciences, The Ohio State University, Columbus, OH, USA. [10]Department of Neurological Surgery, The Ohio State University, Columbus, OH, USA. [11]Department of Neurological Surgery, Nationwide Children's Hospital, Columbus, OH, USA. [12]These authors contributed equally: Toni S. Pearson, Nalin Gupta. ✉email: Krzysztof.Bankiewicz@osumc.edu

