## [Peer Review File · Nature Communications]

REVIEWERS' COMMENTS

Reviewer #3 (Remarks to the Author):

Thank you for responding to the reviewer comments. Consider using a reference about the safety of the AAV2 vector (and the concentration used, even if it is delivered to the brain) to address Reviewer 2's comments.

Some additional comments

Introduction

This is a good review of the available studies and of the rationale for this study.

Results

A clear explanation for the study findings.

Line 193-194 Do the authors have an explanation for the persistent elevation of 3-OMD after the gene therapy (i.e., was the gene therapy effective but not enough that it ameliorated the effect of the AADC deficiency and suggests that 100% gene function is not needed for improvement in this population, perhaps reflecting the redundancy of the systems that protect as one gets older)?

Line 275-276 For the reader who does not know, consider explaining briefly (perhaps in the Discussion) why was there a transient worsening in irritability and sleep and appearance of dyskinesia after the gene delivery.

Discussion

Line 313-315 A question – did the families comment on which improvement was thought to have the greatest impact, motor or OGC/behavioral/sleep? In many clinical situations, improvement in sleep/behavior are very much appreciated by caregivers. If they made any comments, is it worthwhile to inform the reader?

Supplementary Table S1 – It may be worthwhile to put a comment at the bottom to inform the reader why data on Subjects 1, 6, 7 were not acquired.

Reviewer #4 (Remarks to the Author):

The authors present a well-designed and well-executed study of targeted AAV2-AADC gene replacement for AADC deficiency. The evidence is convincing that midbrain delivery is associated with improvement in motor function and prevention of oculogyric crises. I have no major comments.

In terms of the previous reviewer's concerns regarding small sample size, and uncontrolled trial design, I agree these are limitations. However, in an ultra-rare disease such as AADC, I believe this is the most appropriate study design, and it standard for similar disorders. The authors clearly reference the robust natural history data that exists for this disorder, and serves as an appropriate control.

I agree with many of the minor comments put forward by the original reviewers, but feel they have been appropriately addressed.

REVIEWERS' COMMENTS

Reviewer #3 (Remarks to the Author):

Thank you for responding to the reviewer comments. Consider using a reference about the safety of the AAV2 vector (and the concentration used, even if it is delivered to the brain) to address Reviewer 2's comments.

Reference 30 is listed that describes long-term safety and expression after intracranial delivery of AAV-AADC vector in patients.

Some additional comments

Introduction

This is a good review of the available studies and of the rationale for this study.

Results

A clear explanation for the study findings.

Line 193-194 Do the authors have an explanation for the persistent elevation of 3-OMD after the gene therapy (i.e., was the gene therapy effective but not enough that it ameliorated the effect of the AADC deficiency and suggests that 100% gene function is not needed for improvement in this population, perhaps reflecting the redundancy of the systems that protect as one gets older)?

The persistent elevation of 3-OMD would be consistent with residual deficiency of AADC function, and we agree that this point deserves additional comment, so have added the following 2 sentences to the Discussion of p. 15: "We did observe a persistent elevation of 3-OMD after gene delivery in all subjects, which is consistent with some degree of residual AADC deficiency. The marked clinical improvements we observed in our subjects therefore did not require complete restoration of AADC function, but the localization of that function, not only the extent, may be a key determinant of clinical outcomes."

Line 275-276 For the reader who does not know, consider explaining briefly (perhaps in the Discussion) why was there a transient worsening in irritability and sleep and appearance of dyskinesia after the gene delivery.

We agree with the reviewer and have now included a sentence on p.11 to explain it: "These motor and behavioral symptoms coincided with the initial AAV-driven expression of AADC and subsequent abrupt increase in dopamine synthesis."

Discussion

Line 313-315 A question – did the families comment on which improvement was thought to have the greatest impact, motor or OGC/behavioral/sleep? In many clinical situations, improvement in sleep/behavior are very much appreciated by caregivers. If they made any comments, is it worthwhile to inform the reader?

Caregivers did mention to the study team that subjects' improvement in sleep and irritability positively impacted the family's sleep and quality of life. We have added the following sentence to the Discussion on p. 13: "Caregivers reported that these changes, particularly the resolution of OGC and improvements in sleep, had a major positive impact on the child's and family's quality of life."

Supplementary Table S1 – It may be worthwhile to put a comment at the bottom to inform the reader why data on Subjects 1, 6, 7 were not acquired.

A comment has been added now to the table to clarify why Subjects 1,6 and 7 were not included. The Vineland-II test was performed 12 and 24 months after gene transfer. Data was not collected for Subject 1, and Subjects 6 and 7 had not yet reached the Month 12 timepoint after surgery at the time of this report (Subject 7 passed away 7 months after surgery).

Reviewer #4 (Remarks to the Author):

The authors present a well-designed and well-executed study of targeted AAV2-AADC gene replacement for AADC deficiency. The evidence is convincing that midbrain delivery is associated with improvement in motor function and prevention of oculogyric crises. I have no major comments.

In terms of the previous reviewer's concerns regarding small sample size, and uncontrolled trial design, I agree these are limitations. However, in an ultra-rare disease such as AADC, I believe this is the most appropriate study design, and it standard for similar disorders. The authors clearly reference the robust natural history data that exists for this disorder, and serves as an appropriate control.

I agree with many of the minor comments put forward by the original reviewers, but feel they have been appropriately addressed.